# Lipidomic Analysis of Cells and Extracellular Vesicles from High- and Low-Metastatic Triple-Negative Breast Cancer

**DOI:** 10.3390/metabo10020067

**Published:** 2020-02-13

**Authors:** Nao Nishida-Aoki, Yoshihiro Izumi, Hiroaki Takeda, Masatomo Takahashi, Takahiro Ochiya, Takeshi Bamba

**Affiliations:** 1Division of Molecular and Cellular Medicine, National Cancer Center Research Institute, Chuo-ku, Tokyo 104-0045, Japan; naoki@fredhutch.org; 2Division of Metabolomics, Medical Institute of Bioregulation, Kyushu University, Higashi-ku, Fukuoka 812-8582, Japan; hiroaki.takeda@riken.jp (H.T.); m-takahashi@bioreg.kyushu-u.ac.jp (M.T.); bamba@bioreg.kyushu-u.ac.jp (T.B.); 3Department of Molecular and Cellular Medicine, Institute of Medical Science, Tokyo Medical University, Shinjuku-ku, Tokyo 160-8402, Japan

**Keywords:** angiogenesis, diacylglycerol, endothelial cells, extracellular vesicles, exosomes, lipidomics, protein kinase D

## Abstract

Extracellular vesicles (EVs) are lipid bilayer nanovesicles secreted from almost all cells including cancer. Cancer-derived EVs contribute to cancer progression and malignancy via educating the surrounding normal cells. In breast cancer, epidemiological and experimental observations indicated that lipids are associated with cancer malignancy. However, lipid compositions of breast cancer EVs and their contributions to cancer progression are unexplored. In this study, we performed a widely targeted quantitative lipidomic analysis in cells and EVs derived from high- and low-metastatic triple-negative breast cancer cell lines, using supercritical fluid chromatography fast-scanning triple-quadrupole mass spectrometry. We demonstrated the differential lipid compositions between EVs and cells of their origin, and between high- and low-metastatic cell lines. Further, we demonstrated EVs from highly metastatic breast cancer accumulated unsaturated diacylglycerols (DGs) compared with EVs from lower-metastatic cells, without increasing the amount in cells. The EVs enriched with DGs could activate the protein kinase D signaling pathway in endothelial cells, which can lead to stimulated angiogenesis. Our results indicate that lipids are selectively loaded into breast cancer EVs to support tumor progression.

## 1. Introduction

Extracellular vesicles (EVs) are lipid bilayer vesicles, with a size of approximately 100 nm in diameter, secreted from most cells. EVs are considered to mediate fundamental cell-cell interaction by delivering cellular functional components, such as miRNAs, DNAs, and proteins [1]. EVs secreted from cancer cells contribute to tumor growth, invasion, and metastasis by transferring signaling molecules to surrounding and distant host cells. EV cargos (RNAs, DNA, and proteins) have been intensively analyzed [2]; however, lipids are the least analyzed although lipids are fundamental components outlining EVs and are suggested to be involved in the EV formation, characteristics, and biological functions [3,4]. Pioneering works on lipid analysis of EVs reported that the lipid composition of EVs is different from that of cells, suggesting a selective loading mechanism for EVs ([5,6,7], reviewed in [4]). Ceramide (Cer), and its generating enzyme, neutral sphingomyelinase2 (nSMase2), contributes to EV formation [8,9]. The activity of phospholipase D2, which produces phosphatidic acid (PA), is also required for EV formation [10,11]. Therefore, further lipid analysis of EVs is desired to provide novel insights into the physiological roles of lipids in EVs.

Lipids are empirically indicated to be correlated with breast cancer risk and development. Obesity is an established risk factor of breast cancer [12], especially in triple-negative breast cancer (TNBC), a malignant subtype which presents lesser therapeutic options than other subtypes [13]. High cholesterol concentrations have been observed in sera from patients with breast cancer [14], although the correlation between cholesterol levels and breast cancer remains controversial [15]. Breast tumors have altered lipid metabolism, and accumulation of products from de novo lipid synthesis correlated with disease stage and cell proliferation [16]. 

Many researchers have reported that EVs from breast cancer including TNBC contribute to its malignancy and progression through delivering miRNAs and proteins [17,18,19,20]. We assumed that the lipid composition of EVs from metastatic TNBC may be altered to contribute to cancer malignancy along with their cargos. However, a comprehensive analysis of lipids from breast cancer EVs is scarce [21], and the biological activities of lipids on EVs are not elucidated. MDA-MB-231 is a widely used human TNBC cell line. Two derivative cell lines, low-metastatic MDA-MB-231-luc-D3H1 (D3H1) and highly lymph node-metastatic MDA-MB-231-luc-D3H2LN (D3H2LN), were originated from primary tumors of mouse orthotopic xenografts and from cells spontaneously metastasized to lymph nodes, respectively. D3H2LN metastasizes to lymph nodes and lungs more frequently compared with D3H1 [22].

Several lipid species mediate signal transductions. Diacylglycerol (DG) serves as a lipid second messenger in the protein kinase C (PKC) signaling pathway [23,24]. In response to extracellular stimuli, DG is generated by hydrolysis of PIP2, enzymatic reactions from phosphatidylcholine (PC) or PA, and directly interacts with PKC to activate signaling cascade of multiple fundamental biological functions, such as cell proliferation, differentiation, cell survival, and apoptosis [25]. A conventional cancer inducer, phorbol ester, mimics DG to activate the PKC signaling pathway, stimulating cell proliferation to induce carcinogenesis of mouse skin [26]. PKC also stimulates angiogenesis in response to vascular endothelial growth factor (VEGF) [27]. DG stimulates other proteins, including protein kinase D (PKD) [25,28], which was an atypical member of the PKC family (PKCµ), but is now re-classified into the CAMK group [29,30]. DG activates PKD directly and through PKC-dependent phosphorylation, to induce cell growth, survival, motility, protein trafficking and lymphocyte biology [30,31], and of note, plays critical roles in vascular biology and angiogenesis [32,33]. PKD is required for biological responses to VEGFs for angiogenesis-related activities, such as migration, proliferation, and tubulogenesis [34,35,36,37], and for tumor-related angiogenesis [38]. PKCδ is a novel subtype of PKC isoforms activated without Ca^2+^ [26] by being recruited to cellular membrane together with PKD in response to DG [39], and activates downstream signals in pulmonary microvascular endothelial cells [40]. 

Previously, we developed a highly-sensitive quantitative lipidomic analytical system using supercritical fluid chromatography fast-scanning triple-quadrupole mass spectrometry (SFC/QqQMS) [41]. SFC/QqQMS is advantageous over shotgun lipidomics analysis due to less ionization suppression, and therefore enables the detection of wide varieties, including low-abundance lipids, with high resolution to the isomeric forms. In this study, we performed a comprehensive analysis of lipid compositions in EVs and cells derived from high- and low-metastatic TNBC cell lines, D3H2LN and D3H1. We further investigated the bioactivity of DG, an enriched lipid species in EVs from D3H2LN, on PKC signaling pathways in endothelial cells. 

## 2. Results

### 2.1. Characterization of EVs Isolated from High- And Low-Metastatic Breast Cancer Cells

We performed a lipidomic analysis of cells and EVs of two TNBC cell lines, high lymph node-metastatic D3H2LN and low-metastatic D3H1, both derived from the same parental cell line, MDA-MB-231 [22]. A schematic representation of sample collection is shown in Figure 1A. Cells and EVs (originated from the same cell culture) were collected from three independent cultures on separate days for experimental reproducibility. EVs were isolated from the conditioned media by the ultracentrifugation method, as in our previous study [42]. For the background control, a medium without cells was processed in the same way, starting from conditioned medium collection to EV isolation (called as “mock sample”).

EVs isolated through the protocol distributed in particle diameter between 30 nm to 500 nm, with 126 nm and 120 nm at the peak for D3H1 EVs and D3H2LN EVs, respectively (Figure 1B). The particle number and protein amount of the collected EVs are summarized in Appendix A. Vesicular structures of EVs with approximately 100 nm in diameter were validated under an electron microscope (Figure 1C). Immunogold labeling detected CD9 and CD63, which are widely recognized marker proteins of EVs, at the surface of D3H2LN EVs (Figure 1C). Western blot analysis further detected EV marker proteins, CD9, CD63, and CD81, in the collected EV samples (Figure 1D). These results validated the characteristics of EVs collected through this procedure.

### 2.2. SFC/QqQMS-Based Lipidomic Analysis in Cells and EVs Derived from High- And Low-Metastatic TNBC Cell Lines

Widely targeted quantitative lipidomic analyses in cells and EVs from D3H1 and D3H2LN were performed using the SFC/QqQMS method [41]. Each lipid class was separated using SFC with a normal-phase diethylamine-bonded silica column because stationary phase with high polarity recognizes head group of lipids rather than their fatty acyl tails. Thus, all lipid molecules in the same class were eluted at similar retention times. The multiple reaction monitoring (MRM) mode of QqQMS was used to select precursor and product ions. An in-house MRM library containing biological lipid molecules was developed by theoretically calculating the *m/z* of a number of relevant lipids. Because of the co-elution of lipids in the same class, the ion-suppression and/or ion-enhancement effects of the biological matrix can be normalized by adding the appropriate internal standards of 15 lipid classes excluding 4 lipid classes (i.e., alkyl-acyl PC, PC (O) and/or alkenyl-acyl PC, PC (P); alkenyl-acyl PE, PE (P); phosphatidylinositol, PI; and cholesterol) (see Materials and Methods for details). In addition, each lipid class was quantified by summing the constituent lipids in the same class. Therefore, the SFC/QqQMS-based analytical platform enabled us to quantitatively capture the differences in individual lipid classes as well as individual lipid molecular species, including low-abundance lipids [41]. We first performed lipid screening with an equal amount of mixture of all cellular samples as the reference sample and detected a total of 484 lipid species from 19 lipid classes with side-chain variations (Appendix A). Then we performed an analysis of individual cell and EV extracts. All 484 targeted lipid species were detected in cellular extracts and 235 lipid species were above the detection level in, at least, five EV samples out of six (Appendix A). Lipid species detected in EVs were consistent, as the rest of the 249 species were not detected in all six EV samples. In the mock sample, some lipid species, such as fatty acid (FA), cholesterol, cholesteryl ester (CE), and triacylglycerol (TG) were detected (Appendix A). After subtraction of the mock sample from the EVs measurements, 229 lipid species remained positive at least in one EV sample, which were included into further analyses.

The sum of mol quantity of all detected lipid species (called as “total lipids” hereafter) in cells was comparable between two cell lines (Appendix A). The amount of total lipids of EVs normalized by particle number was approximately three times lower in D3H2LN than in D3H1, and was similar when normalized with protein contents (Appendix A). This result indicates that D3H2LN EV particles have lower lipid density, but it remains a possibility that the EV samples from D3H2LN contained non-EV particles such as protein aggregates.

### 2.3. Comparison of Lipid Components between EVs and Their Secreting Cells

The composition of 19 lipid classes based on mol percentages to total lipids was compared between EVs and cells. Lipid compositions of cells and EVs were considerably different in both D3H1 and D3H2LN (Figure 2A–C and Appendix A). As reported previously in a prostate cancer cell line, PC-3 [6], cholesterol, and sphingomyelin (SM) were enriched in EVs compared with that in cells. Cholesterol consisted of approximately 80% in EVs and 50% in cells. Although the SM mol ratio to total lipids detected was low in our study, the SM ratio was higher in EVs than in cells (in D3H1, 0.099%/0.199%; in D3H2LN, 0.171%/0.649%, for cells/EVs). As reported, EVs contained less PC and phosphatidylethanolamine (PE) than their original cells; PC ratio was 14% in both cells and less than 3% in EVs; PE ratio was 12% in cells, but less than 4% in EVs (Figure 2A and Appendix A). Phosphatidylglycerol (PG) and PA were below the detection level in all analyzed species with carbohydrate tail variations in all EV extracts, although both lipid species were well-detected in cells (Figure 2 and Appendix A). This result suggested that PG and PA may be selectively excluded during EV formation.

In addition, differences between the cell lines in lipid compositions of cells versus EVs were observed. PE (P) was higher in D3H2LN EVs than in D3H1 EVs, although the percentage of both lipids was similar in both cells. D3H1 EVs contained more cholesterol than D3H2LN EVs, but their levels in cells remained unchanged (Figure 2A and Appendix A). These results underline the concept that lipids are selectively loaded onto EVs from the cell source. 

### 2.4. Comparison of Lipid Components of EVs from High- and Low-Metastatic TNBC

We compared the absolute mol quantity of lipid classes between D3H1 and D3H2LN in cells and EVs after normalized to particle numbers (Figure 2B,C). In cells, the amount of lysophosphatidylethanolamine (LPE), PA, SM, hexosylceramide (HexCer), (statistically significant: *p* < 0.05), PI, DG and TG (statistically not significant: *p* > 0.05) were higher in D3H2LN, and CE was significantly lower when compared with D3H1. In EVs, DG was enriched to 2-folds in D3H2LN. PC, PC (O) and/or PC (P) (*p* < 0.05), lysophosphatidylcholine (LPC), and cholesterol (*p* < 0.05) were lower in D3H2LN than in D3H1. These differences of lipid amount in D3H1 and D3H2LN cells did not correlate with that in EVs, suggesting lipid classes preferably loaded onto EVs differ among cell lines.

### 2.5. Comparison of Individual Lipid Species of EVs and Cells

Lipid components were further compared between D3H1 and D3H2LN with a resolution of individual lipid species with different fatty acyl tails. Figure 3A shows the fold changes of mol-based percentages of lipids in D3H2LN to D3H1 in cells and EVs with statistical significance. Multiple CE species were accumulated in D3H1 cells (Figure 3A, left). In D3H2LN cells, the amount of several DG species with saturated FA tails, such as DG 14:0‒22:0, was higher than in D3H1 cells. In EVs, DG accumulation in D3H2LN was more prominent, but DG species upregulated in D3H2LN EVs were different from cells; multiple DGs with unsaturated acyl chains were enriched to 2–60-folds in D3H2LN EVs than in D3H1 EVs (Figure 3A, right). The most abundant DG in EVs was DG 18:0–18:1 (0.06% of total lipids) with a 6.1-fold increase from D3H1 EVs. The most significantly upregulated DG in D3H2LN EVs was DG 18:1‒20:2 (0.0006% of total lipids) with a 60-fold increase, while the increase in cells was 1.4-folds. The normalized mol quantities of the detected DAGs are summarized in Figure 3B. The enriched DG species in D3H2LN cells differed from D3H2LN EVs in the length of FA chains and saturation, suggesting selective loading of unsaturated DGs to EVs (Figure 3A,B). Additionally, many of the DG species, including the most enriched DG 14:0‒22:0 in D3H2LN cells, were not detected in both D3H1 and D3H2LN EVs, although all of these were detected in cells (Figure 3A,C). This result further supports the existence of a selective loading system of specific DG species to EVs.

Other lipid species were also compared between D3H1 and D3H2LN in cells and EVs (Appendix A). Overall traits of lipid species were similar between cells and EVs. PE P-16:0‒20:1 was significantly increased in both cells and EVs of D3H2LN compared with D3H1 EVs (Figure 3A). However, several lipids accumulated specifically in EVs. Similar to a previous report on PC-3 prostate cancer cells [6], EVs accumulated lipids with C18:1 at the *sn-2* position: phosphatidylserine (PS) 18:0‒18:1, PE 18:0‒18:1, PE 16:0‒18:1, PE 18:1‒18:1, PC 16:0‒18:1, PI 18:0‒18:1, PI 16:0‒18:1, PI 18:1‒18:1 and DG 18:0‒18:1. PS 18:0‒18:1 is functionally important for the formation of EV lipid bilayers [6], and was the highest among PS lipid class in EVs in our analysis (Appendix A). 

### 2.6. DGs in TNBC EVs Activated PKC Signaling Pathway of HUVECs

Next, we investigated whether enriched DGs in D3H2LN EVs have physiological activities. DG serves as a lipid second messenger in the PKC and PKD signaling pathways [23,24]. Activated PKC and PKD signaling pathways by DGs stimulate cell proliferation and angiogenesis in tumors [23,27,38]. Additionally, one of the well-studied functions of cancer-derived EVs is stimulating angiogenesis; EVs from malignant TNBC cells including D3H2LN stimulate angiogenesis in tumor tissues through their cargos, miRNA, and proteins [9,43,44,45]. We aimed to investigate whether enriched DGs in EVs are additional components that stimulate angiogenesis through PKC and PKD signaling pathways.

To confirm that DG can activate PKC signaling pathways of endothelial cells, 1-Oleoyl-2-acetyl-*sn*-glycerol (OAG), a hydrophilic analog of DG [46,47], was supplemented to HUVEC culture, and the activation of PKC signaling pathways was evaluated by detecting phosphorylated PKC isoforms. OAG strongly induced the phosphorylation of PKD/PKCμ at the activation site Ser744/748 [48] and an auto-phosphorylation site, Ser916, for PKD/PKCμ [49] in HUVECs within 30 min (Appendix A). Additionally, OAG weakly increased the phosphorylation of PKCδ at Thr505. Antibodies recognizing phospho-PKCα/βII (Appendix A), phospho-PKCθ, phospho-ζ/λ (data not shown) remained unchanged in response to OAG. This result suggests that OAG activate mainly PKD/PKCμ, and weakly activate PKCδ.

Then, we examined whether D3H2LN EVs with augmented DGs can induce phosphorylation of PKD/PKCµ and PKCδ. First, EV incorporation into HUVECs was confirmed by fluorescently labeled EVs from D3H1 and D3H2LN with HUVECs (Figure 4A). Within 7 h, the supplemented EVs were accumulated inside the cells and became visible under a confocal microscope. To investigate whether DG loaded onto EVs is capable of activating PKC signaling pathways, phosphorylated PKD/PKCµ and PKCδ in HUVECs supplemented by D3H1 and D3H2LN EVs were detected by immunoblot (Figure 4B). Phosphorylation of PKD/PKCμ at Ser916 in HUVECs was induced as soon as 15 min after supplementing D3H1 EVs and D3H2LN EVs, and remained at low levels until 7 h. The phosphorylation level of PKD/PKCμ was slightly higher and lasted longer in HUVECs treated with D3H2LN EVs than in those treated with D3H1 EVs, but not significantly. PKCδ phosphorylation level was also slightly induced after the supplementation of EVs. Phosphorylation levels of other PKC isoforms remained either unchanged or under detection levels in our experimental condition (data not shown). These results indicate that DGs accumulated in EVs induce PKD/PKCμ phosphorylation and related PKC pathways in HUVECs.

As enrichment of DGs in EVs possibly activates PKC signaling pathways in their original cells with EVs secreted by themselves, to stimulate cell proliferation. Phosphorylation levels of PKD/PKCμ in D3H1 and D3H2LN cells were analyzed by immunoblot (Appendix A). In both D3H1 and D3H2LN cells, phosphorylated PKD/PKCμ was under the detection level with an immunoblot using a highly-sensitive detection reagent, suggesting that the DG-containing EVs secreted into the culture did not induce PKD/PKCμ signaling pathway of the EV-producing themselves. These results indicate that TNBC EVs with enriched DGs are more directed to the surrounding cells like HUVECs than their own to stimulate angiogenesis through PKD/PKCμ signaling pathway.

## 3. Discussion

Lipids, which form the outermost layer of EVs, are important for the formation and biological functions of EVs [3]. In this study, we performed a widely targeted quantitative lipidomic analysis of EVs and cells derived from high-metastatic and low-metastatic TNBC cell lines, D3H2LN and D3H1, with our previously-developed method using SFC/QqQMS [41]. Our analysis successfully identified 484 lipid species from 19 lipid classes in cells and 229 lipid species from 17 lipid species in EVs. A comparison of lipid components between cells and EVs, and between cells with differential metastatic frequency revealed selective enrichment of specific lipid species. Additionally, we found that unsaturated DG species were upregulated in D3H2LN EVs compared with that in D3H1 EVs without increasing the amount in the cells. EVs enriched in DGs were biologically active to induce phosphorylation of PKD/PKCμ and PKCδ in endothelial cells, which leads to stimulation of neoangiogenesis.

Several articles have reported comprehensive quantitative analyses of lipid components of EVs from several cancer cell lines, including prostate cancer, colorectal cancer, glioblastoma, hepatocellular carcinoma, and non-small cell lung cancer patients, as well as non-cancerous cells (oligodendroglial cells, mesenchymal stem cells) [3,5,6,50,51,52,53] (summarized and discussed in a review [4]). However, lipidomics on EVs from breast cancer is little studied [21]. As lipid composition is affected by each cell type, growth conditions, and methods to isolate EVs and lipid analysis [4], it is challenging to compare our results with other reports. However, we could find general features of lipid compositions in EVs.

Early-year reports and a pioneering work of comprehensive lipid analysis of EVs of PC-3 by Llorente et al., have shown that EVs enriched cholesterol and SM and decreased PC compared with the source cells [6]. Our results matched the trends: higher mol% of cholesterol and SM, as well as lower PC in EVs from both D3H1 and D3H2LN. The cholesterol ratio in our study was notably higher than that in other reports; more enrichment in EVs based on mol percentage was also found (Figure 2A). However, we must take into consideration that mol% is calculated based on all the detected lipid molecules, which are different among studies. One of the main differences between Llorente’s PC-3 studies and ours is that we did not target SM with long acyl chains (C24 or higher), which may cause mol% of cholesterol in our analyses higher than the report. This may also be a reason why total SM in our samples had a much lower ratio to the total lipids compared with other reports, although it was still enriched in EVs. The main SM species detected in our study, SM d18:1-16:0 and SM d18:1‒22:0 in both cells and EVs, were also enriched in the PC-3 study [6].

PG and PA were below the detection threshold in EVs with all chain variants in our measurement, although they were detected in cells. PA was reported to be partially involved in EV secretions; inhibition of PLD2, an enzyme removing a head group of PC to generate PA, suppress syntenin-containing EV secretion [10]. In previous lipidomic analyses of EVs, PG and PA were either untested or detected at low amount, at 0.1–0.17 mol%, and 0.1–0.16 mol%, respectively [54]. We assume that PA may be important for forming EVs, but PA is not necessarily loaded to EVs at least in breast cancer cells. Many reports showed relatively similar levels of PE in EVs and cells [54]. We observed a lower PE mol ratio in EVs than cells, but mostly due to high cholesterol levels.

Llorente group discussed in their review that a high amount of CE and TG in EVs is a sign of contaminated lipid droplets and lipoproteins during EV isolation [4]. In our samples, we detected several TGs and CE in EVs, but also from the mock sample at a comparable level (Appendix A). Most of the TGs and CE in our EV samples seem to be derived from medium residuals (probably AlbuMAX I contained in Advanced RPMI1640 medium) or contamination from the environment during sample preparation processes. In addition, the carryover or cross-contamination among samples in SFC/QqQMS analyses was negligible. Therefore, our EV samples have fewer concerns of results being confounded by contaminated lipid droplets and lipoproteins produced from ruptured cells and autophagic vesicles, and carryovers. 

We have shown that unsaturated DG was highly enriched in D3H2LN EVs, and further demonstrated biological activities of the EVs to induce PKD/PKCµ phosphorylation in HUVECs (Figure 4). The upregulation of DGs was observed in the data from a recent lipid analysis of exosome subgroups from MDA-MB-231 lung-metastatic derivative than its parental cell line [21]. Although the report only covers several DG species, this result underpins our findings. The importance of lipids in EVs has been suggested in transferring lipids from one cell to others [55], and EV lipid-mediated signal transductions have been reported [56,57,58]. Our results align with the previous reports in suggesting the bioactivity of DGs in EVs to induce signal activation of the recipient cells. Localization of DGs in cells is highly controlled, as they can induce PKC and PKD signals [59]. Therefore, the increase of DG species in D3H2LN EVs suggests that DGs are actively loaded into EVs, with the purpose of mediating signals to endothelial cells to stimulate angiogenesis. It remains unclear whether DG 18:0‒18:1, the most enriched DG species in EVs from D3H2LN, DG 18:1‒20:2, or other structural variants are important for PKD phosphorylation. Elucidating biological functions of lipid species with different side chains and saturation remains challenging because the strategies to control the production of lipids with specific side chains are lacking. Moreover, we cannot distinguish stimulation by DGs from other components in EVs that activate PKD and angiogenesis; previous reports have shown that multiple components, miRNAs, mRNA, and proteins, of D3H2LN EVs activate angiogenesis [9,43,44,45]. We consider DGs work synergistically with other components of EVs to stimulate angiogenesis.

Phosphorylation of PKD was induced by EVs from D3H1 as well as D3H2LN, although DG content was twice higher in D3H2LN EVs (Figure 4B). We assume it was because of the in vitro experimental condition, adding a high amount of EVs in culture (equivalent to 10 μg of protein). As D3H1 EVs also contain DGs, DGs in EVs from D3H1 may still be able to activate PKD. Under physiological conditions where lower concentrations of EVs are provided, the differential effect on angiogenesis might appear.

DG-mediated PKC activation is involved in many other cancer-related functions, such as cell proliferation and immune reactions, such as activated T cell and B cells, mast cells [60] and phagocytosis of macrophages [61] and neutrophils [62,63]. DG in cancer EVs may contribute to EV-mediated education of other surrounding cells to support tumor progression.

## 4. Materials and Methods 

### 4.1. Cell Lines And Cultures

Triple-negative breast cancer cell lines D3H1 and D3H2LN were purchased from Xenogen (Alameda, CA, USA). Breast cancer cells were cultured in RPMI 1640 (Gibco by Thermo Fisher Scientific, Waltham, MA, USA) supplemented with 10% (*v*/*v*) heat-inactivated FBS (Gibco) and 1% (*v*/*v*) antibiotic-antimycotic (Gibco). Human umbilical vein endothelial cells (HUVECs) and their culture medium, EGM-2 BulletKit, were purchased from Lonza (Basel, Switzerland). All cells were cultured in a 5% CO_2_ humidified atmosphere at 37 °C. 

### 4.2. Sample Preparation for Lipidomic Analysis

Three replicates were prepared from individual frozen cell stock at different dates at passage 4 since the cells were recovered. After 2 days from seeding into 15 cm dishes, at reaching 60–70% confluence, the cells were washed twice with PBS without calcium and magnesium (PBS (-)) and cultured for 2 days in Advanced RPMI medium (Gibco) supplemented with 1% antibiotic-antimycotic and 1% GlutaMAX (Gibco). The conditioned media were collected for EV isolation. The remaining cells in the plates were collected as cellular samples, from 2 plates per replicates into individual tubes. Cells were washed with cold PBS (-) twice and detached by a scraper in suspension with PBS (-). Cells collected by centrifugation were snap-frozen, and stored until analysis. For a negative control of EV collection, culture dishes without cells were processed in the same way.

### 4.3. EV Isolation

EVs were isolated using an ultracentrifugation method with filtration and washing, as described in our previous article [42]. The conditioned medium was centrifuged at 2000× *g* for 10 min at 4 °C, and the supernatant was filtered with a 0.22-μm filter unit (Millipore, Billerica, MA, USA). The filtered medium was then ultracentrifuged at 35,000 rpm (average RCF: 151,263× *g*) for 70 min at 4 °C using SW 41 Ti rotor (Beckman coulter, Brea, CA, USA). The pellet was washed with filtrated PBS (-) once and resuspended in the remaining PBS (-) in the tube. Protein concentration of EVs was measured using a Qubit Protein Assay kit (Molecular Probes from Thermo Fisher Scientific). Particle number of EVs was measured by NanoSight LM10 with NTA2.3 Analytical software (NanoSight, Wiltshire, UK). For lipidomic analysis, EVs suspended in PBS (-) were lyophilized.

To label EVs with fluorescent dye, a PKH67 Green Fluorescent Cell Linker Kit for General Cell Membrane Labeling (Sigma-Aldrich, St. Louis, MO, USA) was used. Purified EVs were incubated with 2 µM PKH67 in filtrated PBS (-) for 5 min at room temperature without light. The EVs were washed with filtrated PBS (-) 3 times on a Vivacon 500 (100,000 MWCO) filter (Sartorius stedim, Goettingen, Germany) to remove unincorporated dye. The stained EVs were resuspended in filtrated PBS (-). For non-EV control, the same volume of filtrated PBS (-) without EVs was processed simultaneously. Nanoparticle tracking analysis confirmed that filtrated PBS (-) used for resuspension of EVs contained significantly low particles compared with EV samples (Less than 1/2500).

### 4.4. Lipid Extraction

Lipids were extracted using the Bligh and Dyer method [64] with minor modifications. Lipids were extracted from cellular pellets (cell samples) or freeze-dried EVs (EV samples), respectively, by 1 mL of extraction solvent (methanol/chloroform/water, 10:5:3, *v*/*v*/*v*) after supplementing 20 µL of internal standard mix A (SM d18:1–17:0 and Cer d18:1–17:0, 5 µM; LPC 17:0, 25 µM; monoacylglycerol (MG) 17:0, DG 12:0–12:0, and TG 17:0–17:0–17:0, 50 µM; HexCer d18:1–12:0, 155 µM; and FA 17:0, LPE 17:1, and PC 17:0–17:0, 500 µM) and 10 µL of internal standard mix B (PE 17:0–17:0, 200 µM; CE 17:0, 250 µM; PG 17:0–17:0, 500 µM; PA 17:0–17:0, 700 µM; and PS 17:0–17:0, 2,000 µM). Four lipid classes, including PC (O) and/or PC (P), PE (P), PI, and cholesterol, could not prepare appropriate internal standards. Therefore, semi-quantitative values for each of the four lipid classes were calculated based on an external calibration comparing the ionization and fragmentation efficiencies of each available standard (PC O-16:0-18:1 and PC P-18:0-20:4, PE P-18:0-20:4, PI 18:0-20:4, and cholesterol) with each alternative internal standard (LPC 17:0, LPE 17:1, PS 17:0-17:0, and CE 17:0) (Appendix A). Samples were mixed vigorously by vortex for 1 min followed by 5 min of sonication. The supernatant (700 µL) centrifugated by 16,000× *g*, 5 min at 4 °C was transferred to clean tubes. After mixing with 195 µL of chloroform and 195 µL of distilled water, the aqueous and organic layers were separated by vortex and the following centrifugation. The bottom layer was diluted into a half by methanol and used for lipidomic analysis. A reference sample (120 µL) was firstly prepared by mixing equal amounts (10 µL each) of 12 TNBC (6 D3H1 and 6 D3H2LN) cell extracts, which was subsequently analyzed using the in-house lipid MRM library [41].

### 4.5. Lipidomic Analysis by SFC/QqQMS

Lipid analysis was performed by SFC/QqQMS in MRM mode as described previously [41]. The SFC/QqQMS system is composed of an ACQUITY UPC^2^ and a Xevo TQ-S micro triple-quadrupole mass spectrometer (Waters, Milford, MA, USA). The SFC conditions were as follows: injection volume, 1 µL; mobile phase (A), supercritical carbon dioxide; mobile phase (B) (modifier) and make-up pump solvent; methanol/water (95/5, *v*/*v*) with 0.1% (*w*/*v*) ammonium acetate; flow rate of mobile phase, 1.0 mL min^‒1^; flow rate of make-up pump, 0.2 mL min^‒1^; modifier gradient; 1% (B) (1 min), 1%‒65% (B) (11 min), 65% (B) (6 min), 65%‒1% (B) (0.1 min), 1% (B) (1.9 min); column, ACQUITY UPC^2TM^ Torus diethylamine (DEA) (3.0 mm × 100 mm, 1.7 µm, Waters); column temperature, 50 °C; active back pressure regulator, 1500 psi; and analytical time, 20 min. The QqQMS conditions were as follows: capillary voltage, 3.0 kV; desolvation temperature, 500 °C; cone gas flow rate, 50 L h^‒1^; and desolvation gas flow rate, 1000 L h^‒1^. The MRM parameters per one time-period were as follows: limit on number of MRM transitions, 150; dwell time, 1 ms; MS inter-scan and inter-channel delay, 2 ms; and polarity switch inter-scan, 15 ms. One cellular sample from each group was used for preliminary experiment. Two cellular samples from the same culture batches were calculated as technical repeats. For the calculation of the lipid amounts in EVs, the mol amount of individual lipid species detected in the mock sample was subtracted from those of EV samples. The negative values were flattened to zero. For comparison between D3H2LN and D3H1, the lipid amount of EVs was normalized by particle numbers.

### 4.6. Supplementation of EVs to HUVECs

HUVECs were seeded at 2 × 10^4^ cells/500 µL/well in 8-well CultureSlides (BD Falcon, Franklin Lakes, NJ, USA) for the uptake assay, or 4 × 10^5^ cells/2 mL/well in 6-well plate for the collection of cell lysate, and incubated for 2 days. For the EV uptake assay, 10 µg of PKH67-stained EVs based on protein amount were supplemented to HUVECs and were incubated for 7 h. For the negative control, PBS (-) without EVs that underwent PKH67 staining was supplemented. Cells were fixed with 4% paraformaldehyde phosphate buffer solution (Wako, Osaka, Japan) for 10 min and mounted with Vecta Shield mounting medium with DAPI (Vector laboratories, Burlingame, CA, USA). EV uptake was observed by confocal fluorescence microscopy (FLUOVIEW FV10i, Olympus, Tokyo, Japan).

For PKC signal stimulation assays, HUVECs were supplemented with 10 µg protein of unstained EVs or PBS (-) as a negative control and incubated for the indicated time. For a positive control of PKC signaling activation, 1-Oleoyl-2-acetyl-*sn*-glycerol (OAG) (Sigma-Aldrich), was supplemented at the final concentration at 50 µM and the same volume of DMSO for vehicle control. After washing with PBS (-), cells were lysed by M-PER Mammalian Protein Extraction Reagent (Pierce from Thermo Fisher Scientific). The supernatants after centrifugation at 14,000× *g*, 5 min of cell lysates were used for protein assays.

### 4.7. Immunoblot

Protein concentration was measured by a micro BCA assay. For electrophoresis the same amount of protein was loaded per well: 2 μg protein for HUVEC lysate of with EVs, 3 µg protein of HUVEC lysate stimulated with OAG, and 2.7 μg protein for D3H2LN and D3H1 cell lysates. For EVs, 8.6 μL of EV suspension per well was used. Cell lysates or EV suspensions were denatured in 4× sample buffer solution with or without 2-ME (Wako), respectively, and separated through standard SDS-PAGE and western blot procedure. Signals were detected using ImmunoStar LD (Wako). Phospho-PKC Antibody Sampler Kit (#9921 from Cell Signaling Technology, Danvers, MA, USA) was used for detection of PKC signaling pathways in HUVECs. For detection of EV markers, anti-CD9 (12A12), anti-CD63 (8A12), anti-CD81 (12C4) (Cosmo Bio, Tokyo, Japan) were used. Anti-tubulin antibody (Cell Signaling Technology, #3873) followed by ECL anti-mouse IgG, HRP-linked whole antibody (from sheep) (GE Healthcare, Chicago, IL, USA), were used for loading control detection. Immuno-electron microscopy of EVs was performed as described in [42].

## 5. Conclusions

We performed a widely targeted quantitative lipidomic analysis of cellular and EVs of high- and low-metastatic TNBC cell lines using SFC/QqQMS, and revealed differential lipid compositions between cells and EVs, which support the idea of selective lipid loading into EVs. Comparison between EVs from high- and low-metastatic cells exhibited high-metastatic D3H2LN EVs were enriched with unsaturated DG species. We further demonstrated the biological activities of DGs enriched in EVs by detecting activated PKD signaling pathway in HUVECs. Our findings explicit the concept that lipids of EVs can mediate intercellular signaling, and they suggest that malignant cancer cells load specific lipids aiming for signal transduction to support cancer growth.

## Figures and Tables

**Figure 1 metabolites-10-00067-f001:**
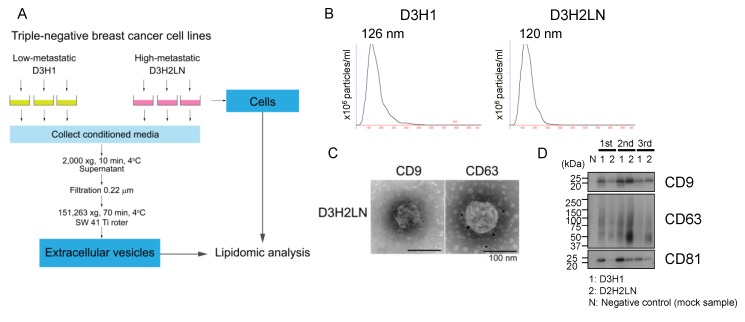
Isolation and characterization of extracellular vesicles (EVs) from high- and low-metastatic breast cancer cell lines. (**A**) Schematic representation of sample preparation. EVs were isolated from conditioned media of low-metastatic breast cancer cell lines, MDA-MB-231-D3H1 (D3H1), and high-metastatic MDA-MB-231-D3H2LN (D3H2LN) by ultracentrifugation with 3 independent replicates. The exact same cells were used for lipidomic analyses. (**B**) Particle distributions of D3H1 and D3H2LN EVs by nanosight particle tracking system. (**C**) Immuno-electron microscopy images of EVs isolated from D3H2LN. CD9 and CD63 at the surface of EVs from D3H2LN were detected by antibodies conjugated with gold nanoparticles. Scale bar: 100 nm. (**D**) Immunoblot of EV marker genes in D3H1 and D3H2LN EVs. Negative control (mock sample) was prepared through the same EV isolation procedure from cell-free control medium.

**Figure 2 metabolites-10-00067-f002:**
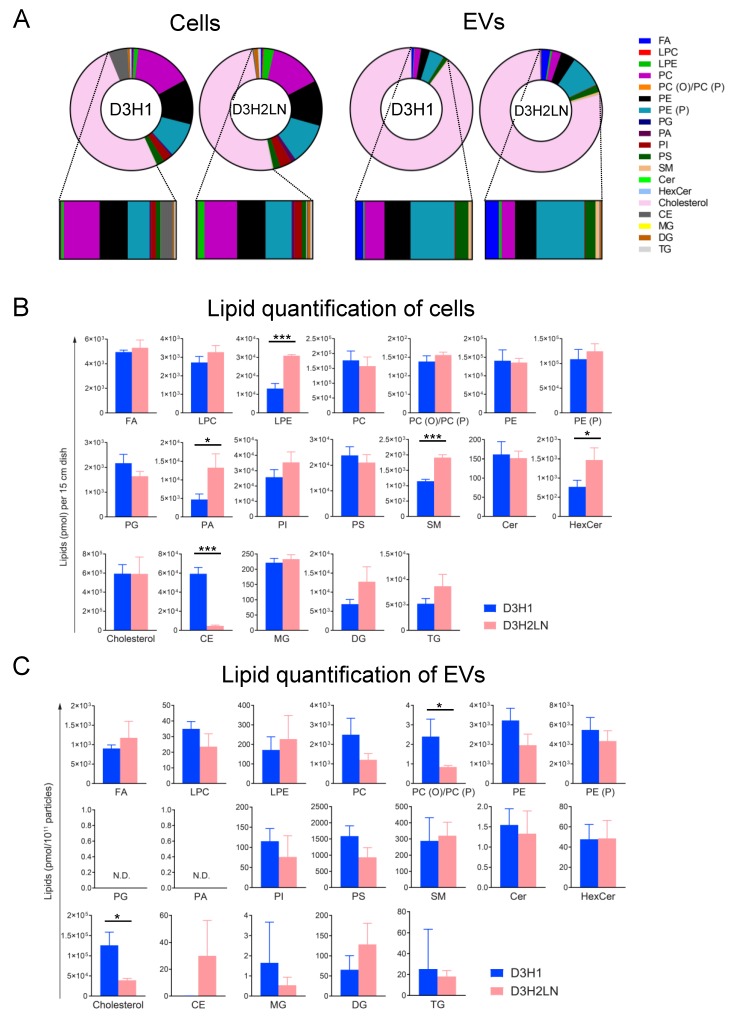
Comprehensive lipid analyses of cellular and EVs of TNBC cell lines. (**A**) Mol-based percentile of each lipid class in cellular and EVs of D3H1 and D3H2LN cell lines. Mol percentage to the sum of all detected lipids was calculated. (**B**) Comparison of the absolute amount of lipid classes between D3H1 and D3H2LN in (**B**) cells and (**C**) EVs (subtracted by mock sample, normalized to particle number). The bar graphs indicate the mean ± SEM of 3 independent replicates. N.D. indicates not detected. *** *p* < 0.001, ** *p* < 0.01, * *p* < 0.05, Student’s t-test (unpaired, two-tailed).

**Figure 3 metabolites-10-00067-f003:**
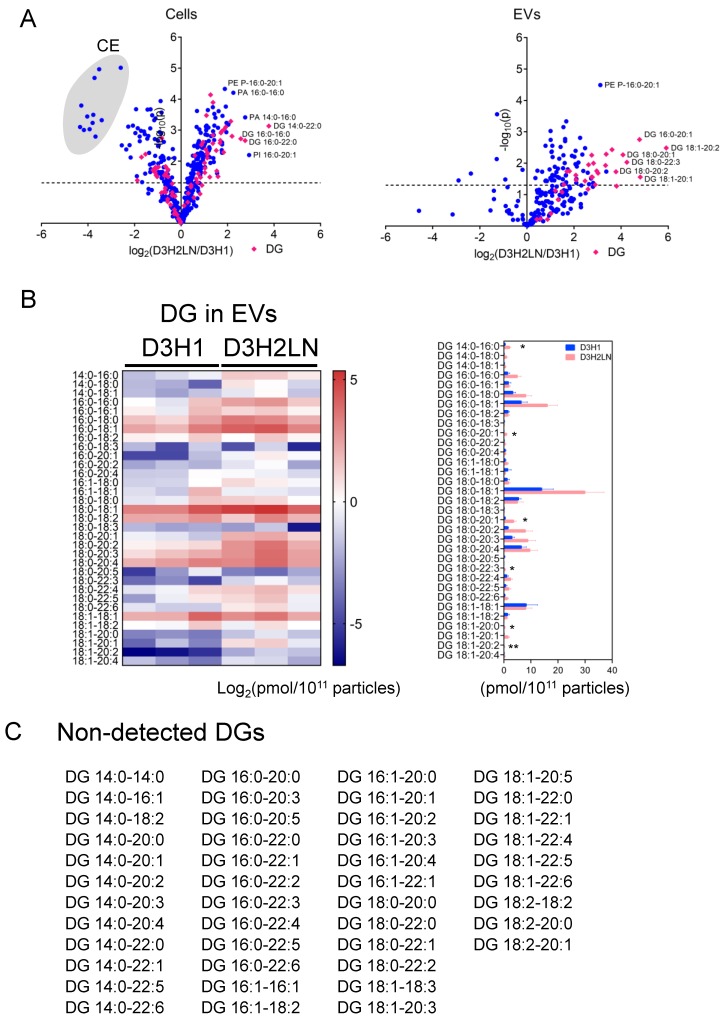
Comparison of individual lipid species between D3H1 and D3H2LN in cells and EVs. (**A**) Volcano plots of fold changes of individual lipid molecules between D3H1 and D3H2LN based on mol percentile to the total detected lipids. Left: cells, Right: EVs. *p*-value was analyzed by an unpaired two-tail Student’s t-test. The plots marked in pink diamonds are DGs. (**B**) (Left) A heatmap of log-based mol quantities of DG species in EVs of D3H1 and D3H2LN, normalized by particle number of EVs. Each column indicates the measurement from three independent samples. (Right) Mol quantity of DGs normalized to particle number of EVs from D3H1 and D3H2LN. The bar graphs indicate mean ± SEM. (C) A list of DG species not detected in both D3H1 and D3H2LN EVs. *** *p* < 0.001, ** *p* < 0.01, * *p* < 0.05, Student’s t-test (unpaired, two-tailed).

**Figure 4 metabolites-10-00067-f004:**
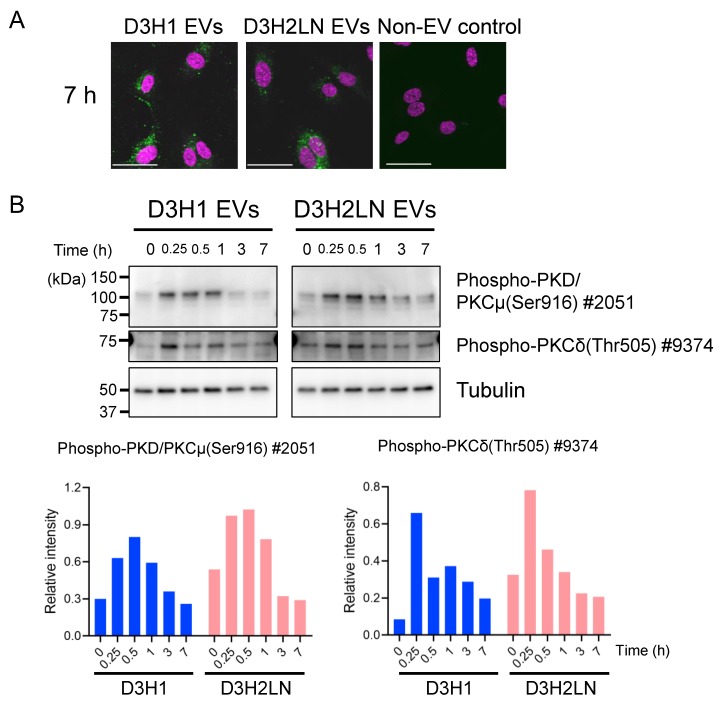
Biological activities of DG accumulated in EVs. (**A**) Microscopic images of HUVECs uptaking fluorescent-labeled EVs. HUVECs were imaged after 7 h from supplementing D3H1 or D3H2LN EVs labeled with green fluorescent dye, PKH67. PBS without EVs that underwent the same fluorescent labeling procedure was supplemented for negative control. Nuclei were labeled with DAPI (shown in magenta). Scale bar: 50 μm. (**B**) (Top) Immunoblot of time-course phosphorylation of PKD/PKCμ in HUVECs after D3H1 or D3H2LN EVs were supplemented. Tubulin was detected as a loading control. (Bottom) Quantification of the intensity of immunoblot bands normalized with that of tubulin.

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
