# Peer review of "Lipidomic Analysis of Cells and Extracellular Vesicles from High- and Low-Metastatic Triple-Negative Breast Cancer"

_metabolites, 2020, doi:10.3390/metabo10020067_

Round 1

Reviewer 1 Report

The manuscript entitled “Lipidomic analysis of cells and extracellular vesicles from high- and low-metastatic triple-negative breast cancer” treats about widely targeted quantitative lipidomic analysis of cells and extracellular vesicles (EV) form high- (D3H2LN) and low-metastatic (D3H1) triple-negative breast cancer. Conducted research fills the gap in the knowledge about lipid content of EV, providing structural and functional insights about lipids forming EV secreted by D3H1 and D3H2LN cells.

I do believe this manuscript is high importance and interest for the readers of Metabolites and I recommend its publication after minor revision: However, there are some issues which must be solved before publishing this work: main concerns are about little number of biological replicates, lack of statistical analysis (or at least lack of any information about statistical analysis performed), lack of the information about data processing and lipids identification. Undeniably, a very strong point of the presented work is the validation of obtained results by measuring the biological activity of DGs and its impact on the PKC signalling pathway. Therefore, I recommend the acceptance of the presented manuscript after some changes. All comments and found issues are listed below:

1. Line 26: “widely targeted quantitative lipidomic analysis” – as far as I understood this method employed single internal standard per lipid class, therefore the method should be rather assumed as semi-quantitative that quantitative.

2. Lines 45-47: Please add a reference for this sentence.

3. Lines 61-64: The contribution of lipid bilayer EVs to TNBC malignancy and the altered lipid composition in breast cancer led us to assume that lipid components of EVs from TNBC are altered, and may contribute to cancer malignancy.” EVs contribute or may contribute to cancer malignancy because this sentence is not clear and provides contradicting information. Please re-write this sentence.

4. Line 71: From results, I understand why DGs are described with more details int the introduction, however, it is not clear when reading the manuscript for the first time. In my opinion, there is no transition or explanation of why exactly DGs are described.

5. Line 100: there is no need to repeat the full the name of the cell lines used (high lymph node-metastatic MDA-MB-231-luc-D3H2LN (D3H2LN) and low-metastatic MDA-MB-231-luc-D3H1 (D3H1)), since this was described in the introduction. Please use directly the high lymph node-metastatic D3H2LN and low-metastatic D3H1.

6. Line 103: Why only 3 cultures were used? What exactly means “different timelines”?

7. The results section should be revised and changed since it contains many repetitions from the methods section. This section should provide results, not information on how these results were generated. Please make adequate changes.

8. Line 135: “A total of 484 lipid species from 19 lipid classes …” Here authors mention 19 lipid classes while in the methods section only 15 lipid classes are covered with used internal standards. Please can you comment on this and how did you quantify missing classes of lipids?

9. Lines 143-144: “After subtraction, 229 lipid species remained positive value at least in 1 EV sample, which was used for further analyses.” I do not know if I understand correctly this sentence: Do this mean that even if the positive value was only in one sample (which means that in 5 other values were negative) the lipid was kept for the further analysis? If so, how did you treat negative values? Did you keep them as negative values or did you replace them by zeros? Please comment on this and clarify this point in the manuscript. However, this section and all explanations should be moved to the method section.

10. Lines 146-147: “Total lipids of EVs normalized to particle number or protein contents, …” Was the normalisation finally performed to the particle number or protein contents? Please clarify this and explain why this particular method was used.

11. Line 165: How did you identified so precisely PE as a (PE(p)) form? How did you eliminate the possibility of the (PE(o)) form? What about PC(o) and PC(p)?

12. Lines 166-168: How your finding of cholesterol content in the cells and EVs relate to these previously published? Please comment on this.

13. Did you perform any statistical analysis? If so, please indicate the statistical significance on the graph in figure 2 (panels B and C) add information about statistical analysis in the Method section. If the statistical analysis was not performed, please conduct such an analysis. Please, indicate clearly in the manuscript, which of the mentioned differences, across all of the comparisons, are statistically significant.

14. Line 184: Add to the heading “and cells” because the heading informs only about EVs while the entire section treats also about the cells.

15. 5. Comparison of individual lipid species of EVs: Usually there is a correlation between intra- and extra- cellular amount of some metabolites, e.g. an increase in the extracellular content correlate with a decrease in the intracellular space and vice versa. Sometimes an increase in the extracellular content is not followed by the decrees in the intracellular content due to the upregulation in intracellular space. However, complete lack of any correlation between intra- (cells) and extra- (EVs) lipid content is quite strange. Can you please comment on this and maybe point analytical aspects which might explain this?

16. Figure 2: Panel A – please change the colours of the graphic because some colours are too similar and the reading of the chart in ambiguous. Please change the colours or add the tags to the chart. Panel B – please provide either better quality figure or increase the axes' font.

17. Lines 219-226: This part should be moved e.g. to the discussion.

18. Lines 272-273: Sentence: “The SFC/QqQMS-based method…..”. This is a repetition.

19. Lines 281-285: Repetition from the introduction.

20. Lines 309-316: Please comment also on the carry-over.

21. Methods: Unify the representation of thousands and represent all the number in the same way, either with or without coma.

22. 4 Lipid extraction: What was the proportion of the cells and EVs and solvents? Please clarify this in the manuscript.

23. 4 Lipid extraction: Did you try to analyse chloroform phase form EVs samples without the dilution? This could help with the detection of low abundant lipids.

24. Lines 402-403: Were QCs prepared only from cells extracts? What about EVs? Did you use another QC for EVs?

25. How the samples were analysed? Did you analyse all the samples (cells and EVs) together within single worklist or in two separate batches?

26. Please revise the references: there is double numeration of references and some references have a different style to all other (e.g. ref 1, 18, 24, 27, 31, 59).

Author Response

Author's Reply to the Review Report (Reviewer 1)

 Comments and Suggestions for Authors

The manuscript entitled “Lipidomic analysis of cells and extracellular vesicles from high- and low-metastatic triple-negative breast cancer” treats about widely targeted quantitative lipidomic analysis of cells and extracellular vesicles (EV) form high- (D3H2LN) and low-metastatic (D3H1) triple-negative breast cancer. Conducted research fills the gap in the knowledge about lipid content of EV, providing structural and functional insights about lipids forming EV secreted by D3H1 and D3H2LN cells.

I do believe this manuscript is high importance and interest for the readers of Metabolites and I recommend its publication after minor revision: However, there are some issues which must be solved before publishing this work: main concerns are about little number of biological replicates, lack of statistical analysis (or at least lack of any information about statistical analysis performed), lack of the information about data processing and lipids identification. Undeniably, a very strong point of the presented work is the validation of obtained results by measuring the biological activity of DGs and its impact on the PKC signalling pathway. Therefore, I recommend the acceptance of the presented manuscript after some changes. All comments and found issues are listed below:

Response

We appreciate your informative suggestions and discussions to greatly improve our manuscript. The following are point-to-point responses to each of your comments. We hope that we could answer all your questions. The revision parts have been indicated with red-colored letters in our revised manuscript (revised manuscript marked for reviewers only).

Comment 1

Line 26: “widely targeted quantitative lipidomic analysis” – as far as I understood this method employed single internal standard per lipid class, therefore the method should be rather assumed as semi-quantitative that quantitative.

Response

Our method utilized in this work was published in reference [41] in the revised manuscript. As we discussed in the previous work, we feel reasonable to call this method “quantitative” lipidomics. Although we agree that this method does not prepare internal standards for every single lipid species, the “absolute quantitative” methods for lipidomics are available nowhere today. Our methods are well comparable and have some advantages over the current widely used “quantitative” lipidomics such as shotgun methods. However, related to discussion in Comment 8, quantification of four lipid classes (PC(O) and/or PC (P), PE(P), PI, and cholesterol) with no direct internal standards are rather semi-quantitative methods.
Please see our revised manuscript at lines 126-136 and 398-403.

Comment 2

Lines 45-47: Please add a reference for this sentence.

Response

The references for the “however, lipids are the least analyzed…” were added as [3,4]. Reference [4] was cited as [6] in the original manuscript.

Please see our revised manuscript at line 46.

Comment 3

Lines 61-64: “The contribution of lipid bilayer EVs to TNBC malignancy and the altered lipid composition in breast cancer led us to assume that lipid components of EVs from TNBC are altered, and may contribute to cancer malignancy.” EVs contribute or may contribute to cancer malignancy because this sentence is not clear and provides contradicting information. Please re-write this sentence.

Response

We revised the sentence as follows.

“We assumed that lipid composition of EVs from TNBC may be altered to contribute to cancer malignancy.”

Please see our revised manuscript at lines 60-61.

Comment 4

Line 71: From results, I understand why DGs are described with more details in the introduction, however, it is not clear when reading the manuscript for the first time. In my opinion, there is no transition or explanation of why exactly DGs are described.

Response

We added a transition sentence at the beginning of the paragraph.

Please see our revised manuscript at line 68.

Comment 5

Line 100: there is no need to repeat the full the name of the cell lines used (high lymph node-metastatic MDA-MB-231-luc-D3H2LN (D3H2LN) and low-metastatic MDA-MB-231-luc-D3H1 (D3H1)), since this was described in the introduction. Please use directly the high lymph node-metastatic D3H2LN and low-metastatic D3H1.

Response

We removed the repetitive full terms of the cell lines in the result section accordingly.

Please see our revised manuscript at line 97.

Comment 6

Line 103: Why only 3 cultures were used? What exactly means “different timelines”?

Response

As our work is on the established cell lines, three repeats are considered to be enough to reveal reliable differences. Cells from the same cell line are artificially similar to each other and biological variations among the trials using the same cell lines in culture are much smaller than primary samples such as patients and laboratory animals.
To obtain more reliable findings which are not confounded by specific experimental conditions, the cell cultures were prepared on different days, which was meant by “different timelines”. We revised the word to “separate days” for clarity, and added words “for experimental reproducibility”.

Please see our revised manuscript at lines 98-100.

Comment 7

The results section should be revised and changed since it contains many repetitions from the methods section. This section should provide results, not information on how these results were generated. Please make adequate changes.

Response

We deleted the detailed description of the procedures, or moved to the method section.
Please see our revised manuscript in Results and Methods sections.

Comment 8

Line 135: “A total of 484 lipid species from 19 lipid classes …” Here authors mention 19 lipid classes while in the methods section only 15 lipid classes are covered with used internal standards. Please can you comment on this and how did you quantify missing classes of lipids?

Response

Thank you for pointing out. We added a description to the revised manuscript according to the experimental procedure we performed.

Please see our revised manuscript at lines 126-136 and 398-403.

Comment 9

Lines 143-144: “After subtraction, 229 lipid species remained positive value at least in 1 EV sample, which was used for further analyses.” I do not know if I understand correctly this sentence: Do this mean that even if the positive value was only in one sample (which means that in 5 other values were negative) the lipid was kept for the further analysis? If so, how did you treat negative values? Did you keep them as negative values or did you replace them by zeros? Please comment on this and clarify this point in the manuscript. However, this section and all explanations should be moved to the method section.

Response

After mock subtraction, at least 1 EV sample out of 6 EVs (D3H2LN and D3H1) was positive. All the negative values are flattened to zero. The brief description of the detected values was included in the Results, and the more detailed version was included into the Method section.

Please see our revised manuscript at lines 146-148 and 425-427.

Comment 10

Lines 146-147: “Total lipids of EVs normalized to particle number or protein contents, …” Was the normalisation finally performed to the particle number or protein contents? Please clarify this and explain why this particular method was used.

Response

We revised the sentence of comparison of sum amount of the detected lipids. The description of the normalization (per particle number) for comparison of lipid groups and individual lipid species was included into Results 2.4 and later. Generally, either particle number or protein amount is used for normalizing EVs. Particle number was used in our manuscript because the particle is the most reasonable unit for counting EVs. In our experiment, total lipid amount of EVs normalized the particle number and protein amount provided comparable values (Fig. S1B), so we suppose either normalization methods will provide similar results in the rest of the data.

Please see our revised manuscript at lines 150-152 and 427-428.

Comment 11

Line 165: How did you identified so precisely PE as a (PE(p)) form? How did you eliminate the possibility of the (PE(o)) form? What about PC(o) and PC(p)?

Response

Our method utilized in this work was published in reference [41] (especially, Supplemental Tables S3 and S5) in the revised manuscript.

Our analytical system using supercritical fluid chromatography triple quadrupole mass spectrometry (SFC/QqQMS) can be identified each lipid molecule based on RT information of lipid class separation by SFC and lipid-specific fragment information by QqQMS in the MRM mode. We identify lipid molecular species based on the following specific MRM transitions and RT information. As you know, in some isomers, PC (O) and PC (P) cannot be distinguished (Table S1). Also, because PE (O) does not have a standard, its fragment information could not be confirmed. Therefore, if PE (O) is present, PE (P) may not be distinguished from PE (O).

PC

MS: [M + CH3COO]

MS/MS: [Acyl FA (sn-1) ‒ H] and [Acyl FA (sn-2) ‒ H]

Alkyl-acyl PC (O) and/or Alkenyl-acyl PC (P)

MS: [M + H]+

MS/MS: [LPC O-FA and/or P-FA (sn-1) + H]+ and [LPC ‒ OH]+

PE

MS: [M ‒ H]

MS/MS: [Acyl FA (sn-1) ‒ H] and [Acyl FA (sn-2) ‒ H]

Alkenyl-acyl PE (P)

MS: [M + H]+

MS/MS: [M ‒ FA ‒ (CH2=COHCH2OH)]+

Comment 12

Lines 166-168: How your finding of cholesterol content in the cells and EVs relate to these previously published? Please comment on this.

Response

We have already discussed cholesterol amount in the Discussion section (lines 292-303).

Comment 13

Did you perform any statistical analysis? If so, please indicate the statistical significance on the graph in figure 2 (panels B and C) add information about statistical analysis in the Method section. If the statistical analysis was not performed, please conduct such an analysis. Please, indicate clearly in the manuscript, which of the mentioned differences, across all of the comparisons, are statistically significant.

Response

For all the bar graphs in this manuscript we calculated statistical significance with Student’s t-test, and showed significance with * marks as follows: *p<0.05, **p<0.01, ***p<0.001 in the revised manuscript. We have included the information in the related figure legends. We also revised the result descriptions to include whether there are any statistical significances.

Comment 14

Line 184: Add to the heading “and cells” because the heading informs only about EVs while the entire section treats also about the cells.

Response

We added the phrase “and cells” to the title of the section 2.5 accordingly.

Please see our revised manuscript at line 190.

Comment 15

2.5. Comparison of individual lipid species of EVs: Usually there is a correlation between intra- and extra- cellular amount of some metabolites, e.g. an increase in the extracellular content correlate with a decrease in the intracellular space and vice versa. Sometimes an increase in the extracellular content is not followed by the decrees in the intracellular content due to the upregulation in intracellular space. However, complete lack of any correlation between intra- (cells) and extra- (EVs) lipid content is quite strange. Can you please comment on this and maybe point analytical aspects which might explain this?

Response

We have shown bar graphs of the amount of individual lipid species in Fig. S3C, where we can compare the trends of the lipids in cellular and EVs. Overall the abundance of each lipid species correlates between cells and EVs, and we agree that generally intracellular and extracellular components do and should correlate each other. However, there are several differences observed as well, which we have discussed in the Results 2.5 section. There considered to be selective cargo loading/excluding systems when forming EVs (ex. Ceramides, as discussed in the Introduction).
One possible explanation of this “mismatch” in regard to the material balance is that the size of EVs is much smaller than cells (100 nm vs 10 µm in diameter, so roughly 100,000 times difference in volume), and the selective loading of certain lipids into EVs does not largely affect intracellular amount of the lipids, even they are secreted thousands per cell.

Comment 16

Figure 2: Panel A – please change the colours of the graphic because some colours are too similar and the reading of the chart in ambiguous. Please change the colours or add the tags to the chart. Panel B – please provide either better quality figure or increase the axes' font.

Response

The panel of the Fig. 2A would be harder to distinguish even with different colors, as there are 19 panels. Therefore, we have added the bar charts of the lipids without cholesterol, to show the ratio of each lipids more clearly. For panel 2B, we replaced to a higher resolution figure.

Comment 17

Lines 219-226: This part should be moved e.g. to the discussion.

Response

We appreciate your suggestion; however, we would like to keep the sentences as in the same location, because we think this short explanation of DG helps readers to understand why we are performing the experiments.

Comment 18

Lines 272-273: Sentence: “The SFC/QqQMS-based method…..”. This is a repetition.

Response

We revised the sentence to remove repetition.

Please see our revised manuscript at lines 277-278.

Comment 19

Lines 281-285: Repetition from the introduction.

Response

We deleted the repetitive sentences.

Comment 20

Lines 309-316: Please comment also on the carry-over.

Response

We added to the sentence.

Please see our revised manuscript at lines 317-318.

Comment 21

Methods: Unify the representation of thousands and represent all the number in the same way, either with or without coma.

Response

We revised the numbers over thousands to represent with comma. We also revised Fig. 1A panel for the comprehensive representation.

Comment 22

4.4. Lipid extraction: What was the proportion of the cells and EVs and solvents? Please clarify this in the manuscript.

Response

Thank you for pointing out. We revised the sentence.

Please see our revised manuscript at lines 391-393.

Comment 23

4.4. Lipid extraction: Did you try to analyse chloroform phase form EVs samples without the dilution? This could help with the detection of low abundant lipids.

Response

We did not try to analyze chloroform phase from EVs samples without the dilution. Next time we aim to detect trace amounts of lipids from EVs samples.

Comment 24

Lines 402-403: Were QCs prepared only from cells extracts? What about EVs? Did you use another QC for EVs?

Response

We changed the QC to a “reference sample” because the expression of the QC was inappropriate.

A reference sample (120 µL) was firstly prepared by mixing equal amounts (10 µL each) of 12 TNBC cell extracts, which was subsequently analyzed using the in-house lipid multiple reaction monitoring (MRM) library. Basically, EV lipids were released from the cells, thus only the cell extracts were used as the reference sample.

Please see our revised manuscript at lines 407-409.

Comment 25

How the samples were analysed? Did you analyse all the samples (cells and EVs) together within single worklist or in two separate batches?

Response

In this study, we analyzed all samples in a single worklist.

Comment 26

Please revise the references: there is double numeration of references and some references have a different style to all other (e.g. ref 1, 18, 24, 27, 31, 59).

Response

We revised the reference duplication and styles, and carefully checked whether it matches the requirements.

The authors would be grateful if you consider this version of manuscript is suitable for publication in Metabolites

Thank you again for your important comments.

Sincerely,

Yoshihiro Izumi

Takahiro Ochiya

Yoshihiro Izumi

Associate Professor

Division of Metabolomics

Medical Institute of Bioregulation

Kyushu University

Email: izumi@bioreg.kyushu-u.ac.jp

Takahiro Ochiya

Professor

Division of Molecular and Cellular Medicine,

Institute of Medical Science,

Tokyo Medical University

Email: tochiya@tokyo-med.ac.jp

Reviewer 2 Report

The authors have presented the work well and the project is interesting. I have few comments

What about the levels of PIP2 in the samples you analyzed? In figure 1D show the band corresponding to CD63. Is there any specific reason to compare Low metastatic c ell lines with High metastatic cell lines, instead of using Healthy or non-cancerous cell lines as control? In all the figures in the manuscript, please mention the significant differences with * or #. In figure 2B, please normalize the data to number of cells or to amount of protein but not to the 15cm dish. Is the expression of enzymes involved in the synthesis of DAG, for example MGAT1, PPH-1/PAP altered in cells and EV? In line 219 correct the word physiologically. In fig 4B, show the bar graph with the quantification of protein bands and significance.

Author Response

Author's Reply to the Review Report (Reviewer 2)

Comments and Suggestions for Authors

The authors have presented the work well and the project is interesting. I have few comments

Response

Authors thank the reviewer for the helpful comments to improve our manuscript. According to your comments and suggestions, the revised points were described below. The revision parts have been indicated with red-colored letters in our revised manuscript marked for reviewers only.

Comment 1

What about the levels of PIP2 in the samples you analyzed?

Response

Unfortunately, we do not have information of PIP2 level of the cells and EVs of the D3H1 and D3H2LN cell lines. We are interested in the level of PIP2 as a precursor of DG, but we currently do not have experimental setups to quantify PIP2 levels. However, PIP2 seems to be important for focal adhesion formation of MDA-MB-231, which stimulates tumor invasion [1]. Additionally, PIPs are involved in vesicular transport and membrane dynamics, possibly EV secretion [2]. Therefore, PIP2 and related DG is possibly play important roles for formation and function of cancer EVs.

(References for the responses)

Fukumoto, M.; Ijuin, T.; Takenawa, T. PI(3,4)P 2 plays critical roles in the regulation of focal adhesion dynamics of MDA-MB-231 breast cancer cells. Cancer Sci. 2017, 108, 941–951. Skotland, T.; Hessvik, N.P.; Sandvig, K.; Llorente, A. Exosomal lipid composition and the role of ether lipids and phosphoinositides in exosome biology. J. Lipid Res. 2019, 60, 9–18.

Comment 2

In figure 1D show the band corresponding to CD63.

Response

The molecular weight of CD63 is 25.6 kDa. However, because CD63 is a heterogeneously glycosylated protein in MDA-MB-231 [3] (our previous work) and other cells [4,5], the WB shows smear band patterns. The smear band patterns were observed in other publications and other suppliers of antibodies of different clones.

(References for the responses)

Tominaga, N.; Hagiwara, K.; Kosaka, N.; Honma, K.; Nakagama, H.; Ochiya, T. RPN2-mediated glycosylation of tetraspanin CD63 regulates breast cancer cell malignancy. Mol. Cancer 2014, 13, 134. Gomes, J.; Gomes-Alves, P.; Carvalho, S.B.; Peixoto, C.; Alves, P.M.; Altevogt, P.; Costa, J. Extracellular vesicles from ovarian carcinoma cells display specific glycosignatures. Biomolecules 2015, 5, 1741–1761. Engering, A.; Kuhn, L.; Fluitsma, D.; Hoefsmit, E.; Pieters, J. Differential post-translational modification of CD63 molecules during maturation of human dendritic cells. Eur. J. Biochem. 2003, 270, 2412–2420.

(CD63 Ab suppliers web pages)

https://www.scbt.com/p/cd63-antibody-mx-49-129-5

https://www.abcam.com/cd63-antibody-epr5702-ab134045.html

Comment 3

Is there any specific reason to compare Low metastatic c ell lines with High metastatic cell lines, instead of using Healthy or non-cancerous cell lines as control?

Response

Thank you for raising the point. We are also interested in lipid composition of EVs from normal breast epithelial cells and would like to compare with those of cancer cells. However, as each cell line drastically alters its characteristics from primary cells during establishment, and also changes over time in the following expansions (MDA-MB-231 was first established in 1970s, and MCF10a in 1984), we concerned the comparison of cancerous and normal cell lines (like MDA-MB-231 and MCF10A) might be confounded by acquired differences during cell lines themselves. For our main focus in this article, to identify lipids involved in cancer malignancy, we thought that comparing the cell lines with the same origin with differential metastatic traits would be the best approach to provide biologically meaningful results with less concerns of introduced difference of cell lines.

Comment 4

In all the figures in the manuscript, please mention the significant differences with * or #.

Response

For all the bar graphs in this manuscript we calculated statistical significance with Student’s t-test, and showed significance with * marks as follows: *p<0.05, **p<0.01, ***p<0.001. We have included the information in the related figure legends.

Comment 5

In figure 2B, please normalize the data to number of cells or to amount of protein but not to the 15cm dish.

Response

We agree that the normalization with protein amount or cell number would be appropriate ways to normalize among samples. However, we think normalization per dish gives us reasonable results for this case, because the total lipid amount collected per dish were comparable level at this study (Fig. S1B).

Comment 6

Is the expression of enzymes involved in the synthesis of DAG, for example MGAT1, PPH-1/PAP altered in cells and EV?

Response

Unfortunately, we do not have expression data nor protein amount of MGAT1, PPH-1/PAP of cells and EVs of D3H2LN and D3H1. We searched for transcriptomic and proteomic data of D3H2LN and D3H1 in the open data source, but none was available. Measuring DG-related enzymes will provide more biological insights into regulation of DGs in cancer cells and selective loadings to EVs.

Comment 7

In line 219 correct the word physiologically.

Response

We revised the word “physiologically” to “physiological”.

Please see our revised manuscript at line 225.

Comment 8

In fig 4B, show the bar graph with the quantification of protein bands and significance.

Response

We quantified the intensity of the WB bands of phospho-PKD/PKCµ and PKCδ, then normalized by that of tubulin. The bar graphs of relative band intensities to tubulin as a loading control are now included into Fig. 4B of the revised manuscript. Also, the WB of the total PKD/PKCµ #2052 did not add any information to the Figure. Therefore, the WB image was removed from the Figure.

The authors would be grateful if you consider this version of manuscript is suitable for publication in Metabolites

Sincerely,

Yoshihiro Izumi

Takahiro Ochiya

Yoshihiro Izumi

Associate Professor

Division of Metabolomics

Medical Institute of Bioregulation

Kyushu University

Email: izumi@bioreg.kyushu-u.ac.jp

Takahiro Ochiya

Professor

Division of Molecular and Cellular Medicine,

Institute of Medical Science,

Tokyo Medical University

Email: tochiya@tokyo-med.ac.jp

Reviewer 3 Report

This is an interesting contribution describing SFC/MS analysis of cells and extracellular vesicles (EV) from high- and low-metastatic cancers. I have the following recommendations how to enhance the quality of this manuscript before the publication.

Major comments

A/ Experimental design - author compare two types of cancer cells and EV for breast cancer, but they do not include normal breast cell lines as a control despite the fact that MCF10A can be obtained relatively easily. The scientific impact of such comparison would be much higher than only the comparison of high vs. low metastatic cancers with missing normal cell lines. Is there some specific reason, why MCF10A is not included?

B/ Method validation - this manuscript does not contain too much information about the analytical methodology, and only the citation of previous work [40] is reported with the remark that previously published work is used for quantitation. I have studies this original paper reporting the analytical methodology, where I miss some substantial components, such as the full method validation in line with FDA guidelines or other relevant guidelines including the use of quality controls and determination of several important parameters, such as matrix effects, selectivity, accuracy, precision, etc. These parameters are quite important for the verification that the MS based methodology really provides robust analytical data, especially in case of the method where internal standards are not coeluting the lipid from individual lipid classes, where the risk of interference is higher and should be carefully excluded by the full method validation.

Minor comments

1/ Line 72 - this is one of possible ways for DG generation, but not the only one. Please reformulate.

2/ Fig. 2 - I miss the information whether observed differences are statistically significant at the level >0.05.

3/ Nomenclature -  I would recommend to follow the established shorthand notation for lipids published in J. Lipid Res. 54 (2013) 1523.

4/ Conclusions - this text is just shortened repetition of abstract.

Author Response

Author's Reply to the Review Report (Reviewer 3)

 Comments and Suggestions for Authors

This is an interesting contribution describing SFC/MS analysis of cells and extracellular vesicles (EV) from high- and low-metastatic cancers. I have the following recommendations how to enhance the quality of this manuscript before the publication.

Response

Authors thank the reviewer for the encouraging comments to our manuscript. According to your comments and suggestions, the revised points were described below. The revision parts have been indicated with red-colored letters in our revised manuscript marked for reviewers only.

Major comment A

Experimental design - author compare two types of cancer cells and EV for breast cancer, but they do not include normal breast cell lines as a control despite the fact that MCF10A can be obtained relatively easily. The scientific impact of such comparison would be much higher than only the comparison of high vs. low metastatic cancers with missing normal cell lines. Is there some specific reason, why MCF10A is not included?

Response

Thank you for raising the point. We are also interested in lipid composition of EVs from normal breast epithelial cells and would like to compare with those of cancer cells. However, as each cell line drastically alters its characteristics from primary cells during establishment, and also changes over time in the following expansions (MDA-MB-231 was first established in 1970s, and MCF10a in 1984), we concerned the comparison of cancerous and normal cell lines (like MDA-MB-231 and MCF10A) might be confounded by acquired differences during cell lines themselves. For our main focus in this article, to identify lipids involved in cancer malignancy, we thought that comparing the cell lines with the same origin with differential metastatic traits would be the best approach to provide biologically meaningful results with less concerns of introduced difference of cell lines.

Major comment B

Method validation - this manuscript does not contain too much information about the analytical methodology, and only the citation of previous work [40] is reported with the remark that previously published work is used for quantitation. I have studies this original paper reporting the analytical methodology, where I miss some substantial components, such as the full method validation in line with FDA guidelines or other relevant guidelines including the use of quality controls and determination of several important parameters, such as matrix effects, selectivity, accuracy, precision, etc. These parameters are quite important for the verification that the MS based methodology really provides robust analytical data, especially in case of the method where internal standards are not coeluting the lipid from individual lipid classes, where the risk of interference is higher and should be carefully excluded by the full method validation.

Response

SFC/QqQMS analysis with an internal standard-dilution method offers quantitative information for both lipid class and individual lipid molecular species in the same lipid class. Therefore, it can be said that the accuracy of quantitative values of each lipid molecules among samples is high.

We have added a description of our SFC/QqQMS method.

Please see our revised manuscript at lines 126-141.

Minor comment 1

Line 72 - this is one of possible ways for DG generation, but not the only one. Please reformulate.

Response

We revised the sentence to include other generation mechanisms of DG.

Please see our revised manuscript at lines 68-73.

Minor comment 2

Fig. 2 - I miss the information whether observed differences are statistically significant at the level >0.05.

Response

We calculated the statistical significance of Fig. 2 and all the other bar graphs using Student’s t-test, and showed significance with * marks as follows: *p<0.05, **p<0.01, ***p<0.001. We have included the information in the related figure legends.

Minor comment 3

Nomenclature -  I would recommend to follow the established shorthand notation for lipids published in J. Lipid Res. 54 (2013) 1523.

Response

According to the International lipid classification, we revised the abbreviations of lipid classes (i.e., alkyl-acyl PC (O), alkenyl-acyl PC (P), and alkenyl-acyl PE (P)) in our revised manuscript and revised supplementary materials.

Minor comment 4

Conclusions - this text is just shortened repetition of abstract.

Response

We revised the Conclusion to include more specific statements.

Please see our revised manuscript at lines 458-465.

The authors would be grateful if you consider this version of manuscript is suitable for publication in Metabolites.

Sincerely,

Yoshihiro Izumi

Takahiro Ochiya

Yoshihiro Izumi

Associate Professor

Division of Metabolomics

Medical Institute of Bioregulation

Kyushu University

Email: izumi@bioreg.kyushu-u.ac.jp

Takahiro Ochiya

Professor

Division of Molecular and Cellular Medicine,

Institute of Medical Science,

Tokyo Medical University

Email: tochiya@tokyo-med.ac.jp

Round 2

Reviewer 3 Report

Authors answered adequately all my comments.